🔓 | **Open Peer Review** | Virology | Research Article

# Epstein-Barr virus (EBV) infection of endothelial cells via endocytosis is associated with a poor prognosis in nasopharyngeal carcinoma

Xu-lin Chen,[1,2] Zhong-heng Huang,[1] Xiu-han Huang,[1] Xi Yao,[1] Ke-ling Pang,[1,3] Xin-lu He,[1,3] Cui-juan Luo,[1,2] Zheng-bo Wei,[1,3] Ying Xie[1,2,4]

**ABSTRACT**  Epstein-Barr virus (EBV) infection is strongly associated with several malignancies, including nasopharyngeal carcinoma (NPC), Burkitt lymphoma, and certain gastric cancers, though its potential to infect endothelial cells (ECs) and the consequent pathological implications remain poorly understood. This study demonstrates through analysis of 99 NPC clinical samples (primary tumors) that Epstein-Barr virus-encoded small RNAs (EBERs) positivity in ECs significantly correlates with N stage (lymphatic metastasis, $P < 0.05$), M stage (distant metastasis, $P < 0.01$), and advanced clinical stage ($P < 0.01$), while immunological profiling reveals concomitant reductions in B-cell proportions ($P < 0.01$) in patients with EBERs-positive ECs. Through comprehensive *in vitro* modeling employing EBV particle infection, Transwell coculture, and direct-contact systems with EBV-positive Akata and Raji cells, we observed that human lymphatic endothelial cells actively internalize infected lymphocytes, with RT-PCR confirming expression of EBV oncogenes (EBNA1, LMP1, and LMP2)—particularly in direct-contact conditions—alongside significant upregulation of endocytosis-related genes Rab5a and EHD1 and ultrastructural evidence of phagosome formation. These findings collectively suggest that phagocytic uptake of EBV-infected lymphocytes or their components may serve as a primary infection mechanism for ECs, potentially establishing an immunosuppressive microenvironment that contributes to tumor progression and poor clinical outcomes, though the exact molecular pathways require further elucidation.

**IMPORTANCE**  Clinical investigations of NPC specimens identified EBERs-positive endothelial cells as clinically significant biomarkers, demonstrating robust correlations with lymphatic metastasis ($P < 0.05$), distant metastasis ($P < 0.01$), and advanced tumor staging ($P < 0.01$), while immunological profiling of affected patients revealed concomitant reductions in B-cell populations ($P < 0.01$), collectively indicative of systemic immunoregulatory status. Furthermore, integrative experimental approaches incorporating live-cell dynamic imaging, single-cell transcriptional profiling, and ultrastructural electron microscopy provided compelling evidence that endothelial phagocytosis of lymphocytes serves as a principal route for EBV cellular entry and subsequent modulation of the NPC tumor microenvironment.

**KEYWORDS**  EBV, nasopharyngeal carcinoma, endothelial cells, prognosis

N asopharyngeal carcinoma (NPC) represents a unique epithelial malignancy arising from the nasopharyngeal mucosa, distinguished from other head and neck cancers by its distinctive etiology, clinical presentation, and therapeutic approaches (1, 2). While environmental and genetic factors contribute to its pathogenesis, the strong association with Epstein-Barr virus (EBV) infection remains a hallmark feature of NPC (3), with

Address correspondence to Zheng-bo Wei, wzhbo1973@aliyun.com, or Ying Xie, xieying@gxmu.edu.cn.

Xu-lin Chen and Zhong-heng Huang contributed equally to this article. Author order was determined by drawing straws.

The authors declare no conflict of interest.

See the funding table on p. 15.

emerging evidence implicating aberrant nuclear factor-κB (NF-κB) signaling activation in its development (4–6). Beyond NPC, EBV has been etiologically linked to diverse malignancies, including Burkitt's lymphoma (7), Hodgkin's lymphoma (8), gastric cancer (9, 10), and plasma cell lymphoma (11), and has even been associated with certain autoimmune disorders like multiple sclerosis (12). Notably, EBV not only drives primary tumorigenesis but also orchestrates an immunosuppressive tumor microenvironment (13–17) that fosters both local tumor progression and metastatic dissemination across multiple cancer types, including NPC (18–21), EBV-associated intrahepatic cholangiocarcinoma (22) and EBV-associated gastric cancer (23, 24).

The vascular endothelium, composed of specialized flat cells lining blood and lymphatic vessels, serves as a dynamic interface regulating substance exchange between circulation and tissues (25). Within the tumor microenvironment, endothelial cells (ECs) play multifaceted roles that extend beyond their classical barrier function (26). They also actively participate in the process of lymphatic and distant metastasis (26). While EBV predominantly targets B lymphocytes and epithelial cells through well-characterized mechanisms (27), emerging evidence challenges the traditional view of endothelial cell resistance to EBV infection. *In vitro* studies demonstrate EBV's capacity to infect human umbilical vein endothelial cells, promoting cell survival via NF-κB pathway activation (28, 29). Clinical observations further support this phenomenon, with EBV-infected endothelial cells identified in various pathological conditions, including AIDS-related epithelioid hemangiomatosis (30), systemic granulomatous arteritis (31), coronary arteritis (32), and notably, in EBV-associated malignancies such as certain gastric cancer and NPC (33).

Despite these intriguing observations, critical gaps remain in our understanding of EBV-endothelial cell interactions, particularly regarding their clinical significance in NPC progression and patient outcomes. The functional consequences of EBV-infected endothelial cells within the NPC immune microenvironment and their potential as prognostic biomarkers remain largely unexplored. This study seeks to address these knowledge gaps by systematically investigating the prevalence, clinical correlates, and immunological impact of EBV-positive endothelial cells in NPC, with particular focus on their relationship with disease progression and patient survival outcomes.

## MATERIALS AND METHODS

### Patients and tumor samples

The paraffin-embedded NPC tissue specimens utilized in this investigation comprised 99 primary tumor biopsies from newly diagnosed NPC patients' tumor biopsies, all consecutively collected at the Affiliated Tumor Hospital of Guangxi Medical University between April 2021 and September 2022. Patient inclusion criteria for primary NPC cases required: (i) initial diagnosis and biopsy performed at our institution; (ii) histopathological confirmation of non-keratinizing squamous cell carcinoma; (iii) absence of any prior anticancer therapies (including radiotherapy, chemotherapy, targeted therapy, or immunotherapy) before biopsy acquisition; and (iv) availability of complete clinical data with adequate tissue samples. To maintain analytical objectivity, all pathological specimens were anonymized through the assignment of unique identification codes unlinked to patient demographics, with sequential tissue sections undergoing blinded staining procedures and independent pathological evaluation by pathologists unaware of clinical outcomes. Tissue sections of pathological specimens were acquired and analyzed using a Tissue Gnostics fully automated panoramic scanning system (Thermo Fisher Scientific with Zeiss AXIO Imager.Z2 microscope and TissueFAXS 7.1.119 software).

### Serum sample processing and humoral immunity assessment

Peripheral blood samples were collected from treatment-naïve patients and processed under standardized conditions: samples were allowed to clot at room temperature for 30 minutes, followed by serum separation. The levels of humoral immunomarkers (IgG,

IgM, IgA, C3, and C4) were quantified using commercially available immunoturbidimetric assay kits in accordance with the manufacturers' protocols, with IgG, IgM, and IgA measured using kits from Guangxi Kang bo lai Technology Co., Ltd. (Nanning, China) and C3 and C4 assessed using kits from MedicalSystem Biotechnology Co., Ltd. (Zhejiang, China). The assay was performed using a biochemical analyzer (ADVIA2400) (Siemens Biochemistry, Germany), and the analyte concentrations were determined based on standard curves generated from each assay.

## Cellular immunity profiling by flow cytometry

Whole blood from untreated patients was collected and coincubated with flow antibodies, and then the proportions of total T lymphocytes, helper lymphocytes, suppressor lymphocytes, natural killer (NK) cells, and B lymphocytes were measured by flow cytometry (Beckman, USA). The kit was purchased from Beckman Coulter.

## Hematoxylin and eosin staining and *in situ* hybridization assay

Hematoxylin and eosin (H&E) staining and EBV-specific *in situ* hybridization (ISH) were performed on NPC tissue sections following standardized protocols (33, 34) to evaluate vascular architecture and EBV infection status. Tissue processing involved baking, sequential deparaffinization, and rehydration through a graded ethanol series. After hematoxylin nuclear staining with acidic alcohol differentiation and ammonia water bluing, sections underwent pepsin digestion to expose nucleic acid targets, followed by ethanol dehydration and air-drying. For Epstein-Barr virus-encoded small RNA s (EBERs) detection, sections were hybridized with digoxigenin-labeled EBERs probes (30 µL/section) in a humidified chamber, then incubated with horseradish peroxidase-conjugated second antibody (1:200, 30 min), followed by color development with 3,3′-diaminobenzidine (Beijing Zhongshan Golden Bridge Biotechnology Co., Ltd., and OriGene Technologies, Inc., Beijing, China).

All stained sections underwent blinded independent evaluation by two senior pathologists using standardized scoring criteria: EBV positivity required strong nuclear staining intensity. Vascular identification relied on characteristic H&E morphological features (endothelial lining and luminal structure), with EBV-infected endothelial cells defined by colocalized EBER signals in ≥3 morphologically confirmed endothelial cells per high-power field (400×). Clinical staging (American Joint Committee on Cancer, 8th edition) and prognostic grading were independently correlated by investigators blinded to molecular results. Discordant cases underwent a third-pathologist adjudication to achieve consensus, ensuring diagnostic accuracy and reproducibility.

## Cells and viruses

Green fluorescence protein (GFP) tagged Akata (Akata-GFP) cells (35, 36) and GFP-tagged Raji (Raji-GFP) cells used in this study were stored and maintained at the Key Laboratory of Early Prevention and Treatment for Regional High-Frequency Tumor, Guangxi Medical University, under standard cell culture conditions. Human lymphatic endothelial cells (HLECs) were purchased from Shanghai Saibaikang Biotechnology Co. Akata-GFP cells were cultured in Roswell Park Memorial Institute (RPMI) 1640 medium supplemented with 9.4% fetal bovine serum and 0.6% Geneticin 418 Solution; Raji cells were cultured in RPMI 1640 medium supplemented with 10% fetal bovine serum, while HLECs were maintained in endothelial cell culture medium (ECM) (ScienCell Research Laboratories) supplemented with 5% fetal bovine serum, 1% endothelial cell growth supplements, and 1% penicillin/streptomycin. All cells were cultured at 37°C in a humidified incubator with 5% $CO_2$ and were passaged using 0.25% trypsin (Gibco).

EBV suspensions were prepared according to a previously published protocol using Akata-GFP cells for EBV production (37). Virus-containing supernatants were collected by ultracentrifugation 48 h after lysis and replication of Akata-GFP cells, which were induced using an anti-IgG antibody at a final concentration of 100 µg/mL. Quantitative analysis of viral DNA was then performed with reference (38) to the literature.

**TABLE 1** Primer sequences

| Gene | Directionality | Sequence (5′–3′) |
|---|---|---|
| LMP1 | Forward primer | GGGGTCGTCATCATCTCCAC |
| | Reverse primer | CATAGCCCTAGCGACTCTGC |
| LMP2A | Forward primer | AATGAAGAGCCCCCACCGCCT |
| | Reverse primer | GCCCGTCATTCCCGTCGTGTT |
| EBNA1 | Forward primer | GGACCCGGCCCACAACCTG |
| | Reverse primer | CTCCTCCCCTTCCTCACCCTCATC |
| CD21 | Forward primer | TGGAACCTGGGATAAACCTGC |
| | Reverse primer | GACTTGTTTCCGTTCATGGAGA |
| Rab5a | Forward primer | GCATGGGTCCCTCTCACTAA |
| | Reverse primer | CAGTGTGGAGAAATGGGCTG |
| Rab7a | Forward primer | CACAATAGGAGCTGACTTTCTGACC |
| | Reverse primer | GTTCCTGTCCTGCTGTGTCCCATATC |
| EHD1 | Forward primer | CCACAAGCTGGACATCTCCGATGAG |
| | Reverse primer | GGGACCAGAAGGAGCCGATGTAGAC |
| ARF6 | Forward primer | ATGGGGAAGGTGCTATCCAAA |
| | Reverse primer | GCAGTCCACTACGAAGATGAGACC |

## Strategic modes of EBV infection in human lymphatic endothelial cells

Three distinct experimental paradigms were employed to investigate EBV infection mechanisms in HLECs: (i) direct viral particle infection (MOI = 1,000), as described in the literature (39); (ii) Transwell-mediated indirect coculture with EBV-positive lymphoma cells (Akata/Raji), which were induced into lytic phase using TPA (20 ng/mL) and sodium butyrate (3 mM); and (iii) direct cellular interaction with EBV-positive lymphoma cells. For direct viral infection, confluent HLEC monolayers ($1.5 \times 10^5$ cells/well, 80% confluence) were exposed to ice-thawed EBV stocks in complete ECM, with gentle agitation to ensure even viral distribution, followed by 72 h incubation, phosphate buffer solution (PBS) washing, and RZ lysis buffer treatment for RNA extraction using RNAsimple Total RNA Kit. The Transwell system utilized Akata-GFP cells ($2 \times 10^5$) in lytic phase (upper chamber) and HLECs (lower chamber), with medium replacement at 48/72 h before lysis. For direct cellular interaction, lytically induced Akata-GFP cells were cocultured with HLECs for 72 h, followed by complement-dependent cytotoxicity to eliminate residual lymphoma cells, according to the protocol indicated in the report of Duensing and Watson (40). Moreover, subsequent RT-qPCR validation of CD21 was carried out to confirm effective depletion prior to RNA extraction from HLEC lysates. All infection conditions included triplicate biological replicates and EBV-negative controls, with lysates stored at $-80°C$ for downstream molecular analyses.

## Real-time fluorescence imaging of Akata cell-HLEC/Raji cell-HLEC interactions

Akata-GFP cells and Raji cells were induced into the lytic replication phase, rigorously washed (3× PBS) to eliminate induction reagents, and cocultured with HLECs at 37°C, 5% $CO_2$ for live-cell imaging using the BioStation IM-Q workstation, with simultaneous brightfield and GFP fluorescence image acquisition initiated immediately upon coculture and captured at 10 minute intervals to dynamically monitor cellular interactions and potential EBV transfer events.

## RT-qPCR detection of endocytosis- and EBV-related genes expressed in HLECs

HLECs in each group experienced the above EBV infection modes and were subsequently used for endocytosis-related genes and EBV-associated gene detection, respectively, by RT-qPCR. In brief, total RNA was extracted from HLECs in each group using Trizol reagent. The purity and concentration of RNA were measured by a Nanodrop 2000 spectrophotometer. cDNA was synthesized from total RNA using reverse transcription, and GAPDH

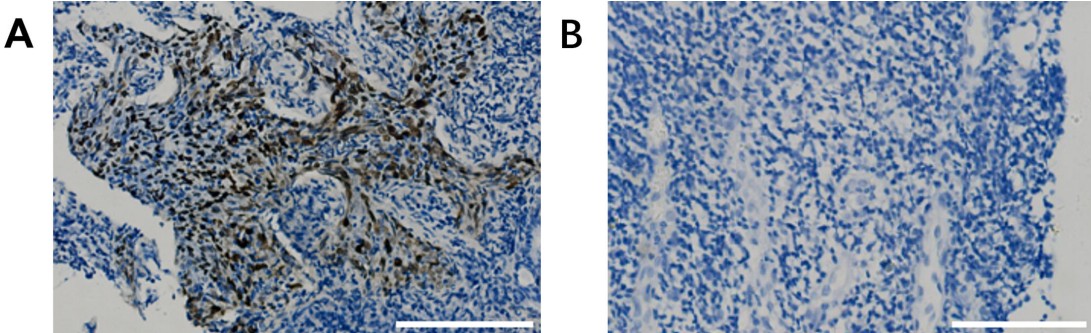

**FIG 1** EBERs-ISH staining of pathological sections of nasopharyngeal carcinoma patients (×200, scale bar = 50 µm). (A) Representative image of EBERs-positive samples. (B) Representative image of EBERs-negative samples.

was used as an internal control for RNA. RT-qPCR was performed following the manufacturer's instructions. The reaction conditions are indicated as follows: predenaturation at 95℃ for 30 seconds to activate DNA polymerase, followed by 40 cycles of 95℃ for 5 seconds and 60℃ for 30 seconds. Melting curve analysis was generated to verify the specificity of the amplification products. After the reaction, the melting curve was analyzed, with the Ct values being recorded. Using the $2^{(-\Delta\Delta Ct)}$ method, the relative expression levels of genes were calculated. The primer sequences for LMP1, LMP2A, EBNA1, CD21, Rab5a, Rab7a, EHD1, and ARF6 are indicated in Table 1.

## Sample preparation for transmission electron microscopy

Following 48 h coculture with Raji cells, HLECs were enzymatically dissociated and processed for ultrastructural analysis through sequential fixation, dehydration, and embedding steps. Primary fixation was performed using 3% glutaraldehyde in 0.1 M PBS for ≥2 h at 4℃, followed by three PBS washes (10–15 minutes each). Secondary fixation employed 1% osmium tetroxide in PBS for 1–2 h at 4℃ with subsequent PBS rinses. A graded dehydration series was conducted starting with ice-cold 50% and 70% ethanol (15 minutes each) at 4℃, progressing through a 1:1 mixture of 90% ethanol and 90% acetone, and three changes of 100% acetone (15 minutes each) at room temperature. Samples were gradually infiltrated with a mixture of acetone and resin (1:3) overnight before final embedding in Epoxy Resin 618 and stepwise polymerization (40℃/15 h → 48℃/12 h → 60℃/48 h). Ultrathin sections collected using an ultrathin slicer (Leica EM UC, Germany) were contrast-stained with uranyl acetate and lead citrate prior to imaging using a transmission electron microscope.

## Obtaining and processing single-cell transcriptome data

The single-cell expression matrix, barcode, and gene annotation for GSE120926 were downloaded from the GEO database (https://www.ncbi.nlm.nih.gov/geo/query/acc.cgi?acc=GSE120926). Seurat v.5.1.0 (41) was used for data processing, with cell filtering based on two criteria: (ⅰ) fewer than 500 detected genes and (ⅱ) mitochondrial gene expression exceeding 20%. Following filtration, the data were processed using Seurat's standard workflow, including the calculation of 20 principal components (via principal component analysis) and anchor-based integration. Cell clustering was performed with a resolution of 0.2, and clusters were identified based on marker gene expression. Further sub-classification of cell types was conducted using the same resolution, with immune cell annotations manually refined using SingleR as a reference (42).

## Acquisition and processing of spatial transcriptome data

The stereo-seq single-cell expression matrix and spatial location data for GSE206245 were obtained from the GEO database (https://www.ncbi.nlm.nih.gov/geo/query/

**TABLE 2** Correlation between clinical characteristics of NPC patients and EBERs status in pathological specimens (n [%])

| Characteristics | n | EBERs Negative(-) n=6 (6.1%) | EBERs Positive(+) n=93 (93.9%) | $\chi^2$ | P |
|---|---|---|---|---|---|
| Age group (years) | | | | – | 1.000 |
| ≤60 | 84 | 5 (6.0) | 79 (94.0) | | |
| >60 | 15 | 1 (6.7) | 14 (93.3) | | |
| Gender | | | | 0.000 | 1.000 |
| Male | 69 | 4 (5.8) | 65 (94.2) | | |
| Female | 30 | 2 (6.7) | 28 (93.3) | | |
| T stage | | | | 1.528 | 0.570 |
| T1 | 0 | 0 (0.0) | 0 (0.0) | | |
| T2 | 32 | 1 (3.1) | 31 (96.9) | | |
| T3 | 39 | 2 (5.1) | 37 (94.9) | | |
| T4 | 28 | 3 (10.7) | 25 (89.3) | | |
| N stage | | | | 2.267 | 0.476 |
| N0 | 7 | 1 (14.3) | 6 (85.7) | | |
| N1 | 28 | 2 (7.1) | 26 (92.9) | | |
| N2 | 29 | 2 (6.9) | 27 (93.1) | | |
| N3 | 35 | 1 (2.9) | 34 (97.1) | | |
| M stage | | | | 0.000 | 1.000 |
| M0 | 83 | 5 (6.0) | 78 (94.0) | | |
| M1 | 16 | 1 (6.3) | 15 (93.7) | | |
| Clinical stage | | | | 0.655 | 1.000 |
| I | 0 | 0 (0.0) | 0 (0.0) | | |
| II | 11 | 0 (0.0) | 11 (100.0) | | |
| III | 29 | 2 (6.9) | 27 (93.1) | | |
| IV A | 43 | 3 (7.0) | 40 (93.0) | | |
| IV B | 16 | 1 (6.3) | 15 (93.8) | | |

acc.cgi?acc=GSE206245). The expression matrix was normalized using Seurat's SCTransform function. Subsequently, annotated single-cell transcriptome data were leveraged to annotate the spatial transcriptome data using the TransferData function. EBV gene expression levels were quantified for each cell, with cells classified as EBV positive if the expression was more significant than zero or EBV negative otherwise. Finally, the proportion of EBV-infected cells was calculated.

## Statistical analysis

All statistical analyses were performed using SPSS v.17.0 (IBM, USA). All data were analyzed using Student's t-test. A P value of <0.05 was considered to indicate a statistically significant difference. Count data were presented as frequency and percentage and analyzed by $\chi^2$ analysis or Fisher's exact test. Measurement data were presented as the mean ± standard deviation, and data with normal distribution were analyzed by an F-test or t-test. Data with uncommon distribution were analyzed by a non-parametric test or Wilcoxon's rank test. Survival curves and between-group comparisons were made using the Kaplan-Meier method for estimating the survival function and the log-rank test (Mantel-Cox test) for comparing between-group differences.

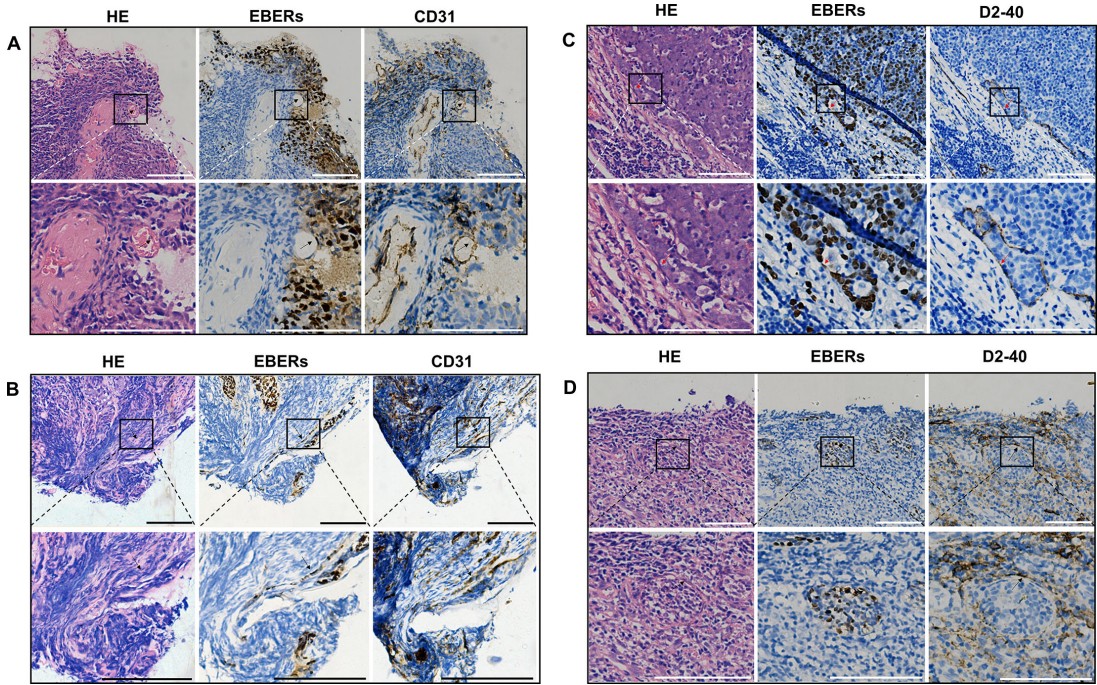

**FIG 2** Endothelial cells in NPC tissues that are positive for EBERs by ISH staining (×200, scale bar = 50 µm). (A and B) Representative images and local magnified views of EBERs-positive vascular endothelial cells in the primary tumor. (C and D) Representative images and local magnified views of EBERs-positive lymphatic endothelial cells in the primary tumors. HE, hematoxylin and eosin; ISH, *in situ* hybridization; NPC, nasopharyngeal carcinoma.

## RESULTS

### Association between EBV infection status and clinical characteristics and prognostic staging in NPC patients

Among the 99 tumor biopsy samples collected from NPC patients, 93 (93.9%) were EBERs positive, while 6 (6.1%) were EBERs negative (Fig. 1). No significant differences were observed between EBERs-positive and EBERs-negative patients regarding age, gender distribution, tumor, node, metastasis (TNM) classification, or clinical stage (Table 2).

### Association between endothelial EBERs positivity and clinical stage of nasopharyngeal carcinoma

ISH analysis of EBERs status in endothelial cells in 99 NPC patients revealed 81 cases (81.8%) with EBERs-negative endothelial cells and 18 cases (18.2%) with EBERs-positive endothelial cells (Fig. 2).

Analysis of the associations between clinical characteristics of NPC patients and EBV infection status in peritumoral vascular endothelial cells revealed statistically significant differences between EBERs-positive and EBERs-negative groups regarding distant metastasis, clinical stage, and overall clinical outcome (Table 3). The results demonstrate that the EBERs-positive group contained a substantially higher proportion of patients with N3 stage (55.56%, 10 out of 18) than that of the EBERs-negative group (30.86%, 25 out of 81). Furthermore, patients in the EBERs-positive group exhibited a significantly greater prevalence of metastatic disease (M1 stage) (44.44% vs 9.9%, $P = 0.001$) and experienced significantly higher rates of tumor relapse or disease progression than their counterparts in the EBERs-negative group (55.56% vs 25.93%, $P = 0.014$).

### Survival analysis

This study enrolled 99 patients. Kaplan-Meier survival analysis of the primary NPC cases demonstrated significantly worse overall survival (OS, $P = 0.020$) and distant

**TABLE 3** Correlation between clinical characteristics and endothelial cell EBER positivity in NPC patients (n [%])

| Characteristics | n | EBERs in endothelial cells | | $\chi^2$ | $P^a$ |
| | | Negative(-)<br>n=81<br>(81.82%) | Positive(+)<br>n=18<br>(18.18%) | | |
| --- | --- | --- | --- | --- | --- |
| Age group (years) | | | | 0.315 | 0.574 |
| ≤60 | 84 | 70 (83.3) | 14 (16.7) | | |
| >60 | 15 | 11 (73.3) | 4 (26.7) | | |
| Gender | | | | 0.680 | 0.410 |
| Male | 69 | 55 (79.7) | 14 (20.3) | | |
| Female | 30 | 26 (86.7) | 4 (13.3) | | |
| T stage | | | | 0.410 | 0.814 |
| T1 | 0 | 0 (0.0) | 0 (0.0) | | |
| T2 | 32 | 26 (81.3) | 6 (18.7) | | |
| T3 | 39 | 33 (84.6) | 6 (15.4) | | |
| T4 | 28 | 22 (78.6) | 6 (21.4) | | |
| N stage | | | | 3.929 | 0.047* |
| N0, N1, and N2 | 64 | 56 (87.5) | 8 (12.5) | | |
| N3 | 35 | 25 (71.4) | 10 (28.6) | | |
| M stage | | | | 10.562 | 0.001** |
| M0 | 83 | 73 (88.0) | 10 (12.0) | | |
| M1 | 16 | 8 (50.0) | 8 (50.0) | | |
| Clinical stage | | | | 13.801 | 0.001** |
| I | 0 | 0 (0.0) | 0 (0.0) | | |
| II | 11 | 10 (90.9) | 1 (9.1) | | |
| III | 29 | 28 (96.6) | 1 (3.4) | | |
| IV A | 43 | 35 (81.4) | 8 (18.6) | | |
| IV B | 16 | 8 (50.0) | 8 (50.0) | | |
| Clinical outcomes | | | | 6.011 | 0.014* |
| Recurrence/progression | 31 | 21 (67.7) | 10 (32.3) | | |
| No recurrence/no progression | 68 | 60 (88.2) | 8 (11.8) | | |

$^a$*$P < 0.05$, **$P < 0.01$.

metastasis-free survival (DMFS, $P = 0.004$) in endothelial EBERs-positive patients compared to EBERs-negative cases (Fig. 3A and B). These findings imply that endothelial EBERs positivity may serve as a potential biomarker for poor prognosis, warranting further validation through larger-scale studies.

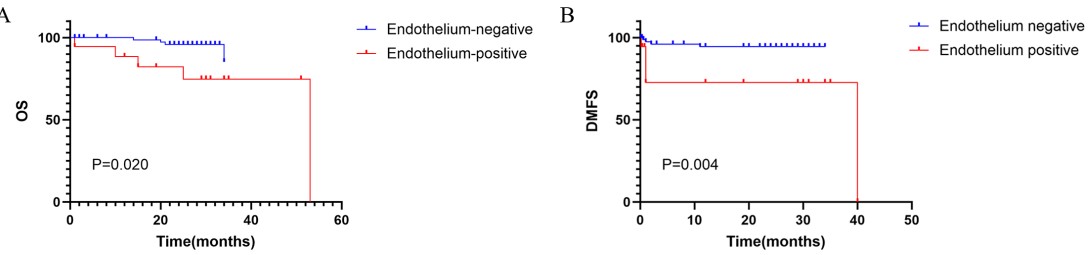

**FIG 3** Survival analysis of 99 primary nasopharyngeal carcinoma patients stratified by endothelial EBERs status. (A) Kaplan-Meier overall survival (OS) curves comparing EBERs-positive (red) and EBERs-negative (blue) groups (log-rank $P = 0.020$). (B) Distant metastasis-free survival (DMFS) curves showing significantly worse outcomes in EBERs-positive patients (log-rank $P = 0.004$).

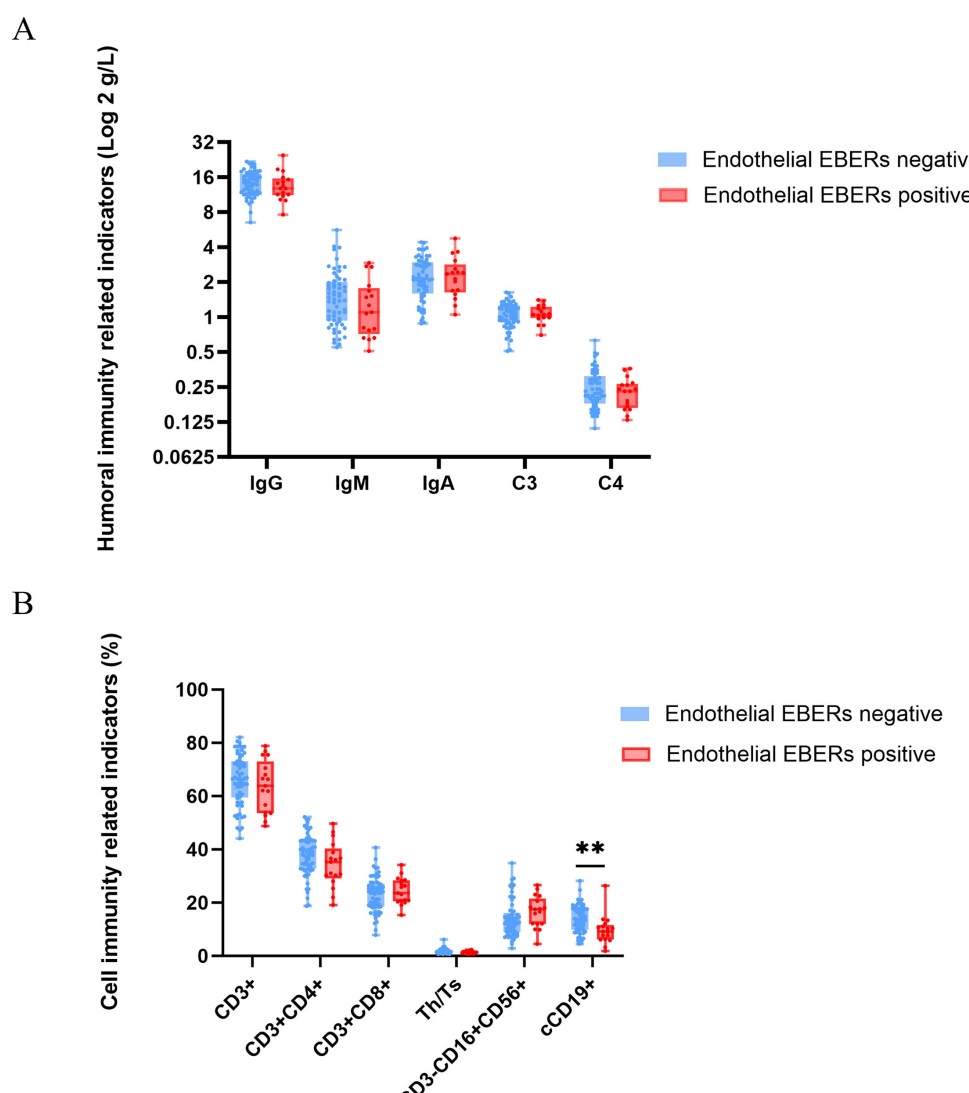

**FIG 4** Box plot showing the indices of humoral immunity and cellular immunity in NPC patients stratified by endothelial EBERs status. (A) Levels of each indicator of serum humoral immunity in nasopharyngeal cancer patients in the EBERs-positive and EBERs-negative groups. (B) Different levels of serum immune cells, including B cells (cCD19+) in nasopharyngeal cancer patients in the EBERs-positive and EBER-negative groups. cCD19, cytoplasmic CD19. **, $P < 0.01$.

## Analysis of humoral and cellular immunity indices in endothelial EBERs-positive NPC patients

Comprehensive serological evaluation of 99 NPC patients demonstrated distinct immunological patterns between EBERs-positive and EBERs-negative cohorts, with the

**TABLE 4** RT-qPCR detection of EBV-related genes and CD21 expression in HLECs upon three different infection modes by EBV from Akata cells[a]

| Gene | Direct viral particle infection | Direct cellular interaction with lymphoma cells[b] | Transwell-mediated indirect coculture with lymphoma cells | Akata cell (positive control) | WT-HLEC |
|------|------|------|------|------|------|
| LMP1 | 0 | 52.13 ± 10.42**** | 0.0047 ± 0.004 | 1 | 0 |
| LMP2A | 0 | 3.885 ± 0.735**** | 0 | 1 | 0 |
| EBNA1 | 0.079 ± 0.059 | 2.789 ± 0.703**** | 0.0344 ± 0.06 | 1 | 0 |
| CD21 | 0 | 0 | 0 | 1 | 0 |

[a]EBV, Epstein-barr virus; WT, wild type.
[b]****, $P < 0.0001$. $P$ values were for the comparisons of EBV-related gene expression profiles between groups exposed to direct Akata cell contact vs those receiving direct viral particle infection or transwell-based mediated indirect coculture with Akata cells.

**TABLE 5** RT-qPCR detection of EBV-related genes and CD21 expression in HLECs upon three different infection modes by EBV from Raji cells[a]

| Gene | Direct viral particle infection | Direct cellular interaction with lymphoma cells | Transwell-mediated indirect coculture with lymphoma cells | Raji cell (positive control) | WT-HLEC |
|---|---|---|---|---|---|
| LMP1 | --/NA | 0.065312 | --/NA | 1 | 0 |
| LM2A | --/NA | 0.043319 | --/NA | 1 | 0 |
| EBNA1 | 0 | 0.009514 | 0 | 1 | 0 |
| CD21 | 0 | 0 | --/NA | 1 | 0 |

[a]EBV, Epstein-barr virus; WT, wild type; --/NA, No expression was detected.

EBERs-positive group showing non-significant trends toward decreased IgG, IgM, and complement C4, along with elevated IgA and C3 levels compared to EBERs-negative patients (Fig. 4A). Flow cytometric analysis of peripheral blood mononuclear cells revealed a marked reduction in circulating B lymphocyte percentage in EBV-positive patients ($P < 0.01$, Fig. 4B), suggesting that EBV infection of the endothelium may promote systemic B-cell depletion through either direct viral effect on lymphoid populations or indirect modulation of humoral immunity, potentially contributing to the observed impairment of tumor immunosurveillance in EBV-driven NPC progression.

### *In vitro* study on the EBV infection pathway in endothelial cells

RT-qPCR analysis of HLECs under three distinct EBV infection strategies demonstrated that untreated wild-type HLECs showed no detectable expression of EBV-associated genes or CD21. Notably, HLECs infected through direct lymphoma cell (Akata/Raji cell) contact exhibited significantly elevated expression of LMP1, LMP2A, and EBNA1 compared to other infection modes ($P < 0.001$, Tables 4 and 5). Complement-dependent cytotoxicity assays confirmed effective elimination of free lymphoma cells, as evidenced by the absence of CD21 expression in directly cocultured HLECs (Table 4). These results establish that EBV may efficiently infect HLECs via B lymphocyte-mediated cell-to-cell transmission, inducing robust expression of viral oncogenes (LMP1, LMP2A, and EBNA1) that may critically influence infected cell proliferation and survival, potentially modulating the NPC tumor microenvironment (Table 5).

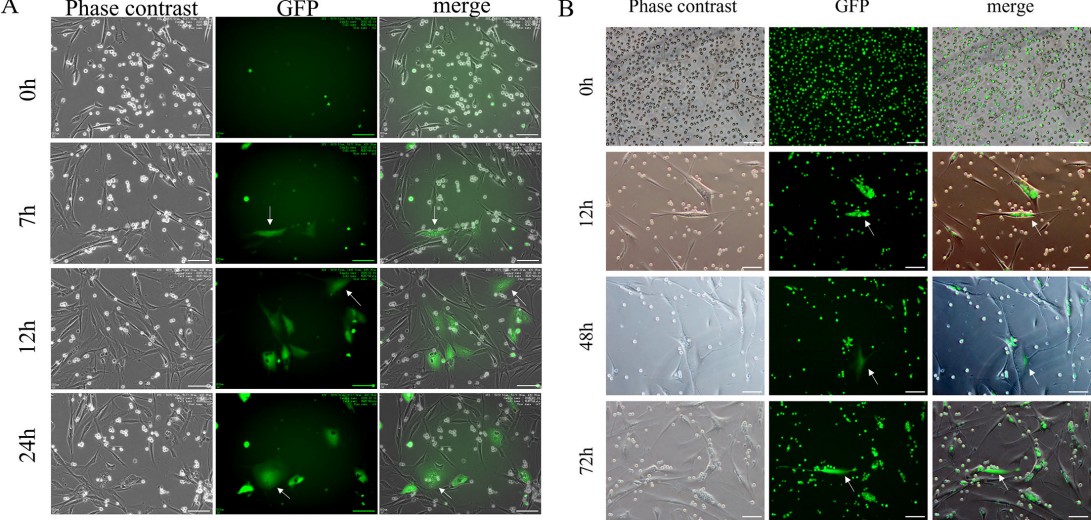

**FIG 5** HLEC directly contacts and internalize Akata-GFP cells and Raji-GFP cells to form "intracellular" structures (×100, scale bar = 100 µm) (A) Coculture of HLEC with Akata-GFP cells. (B) Coculture of HLEC with Raji-GFP cells.

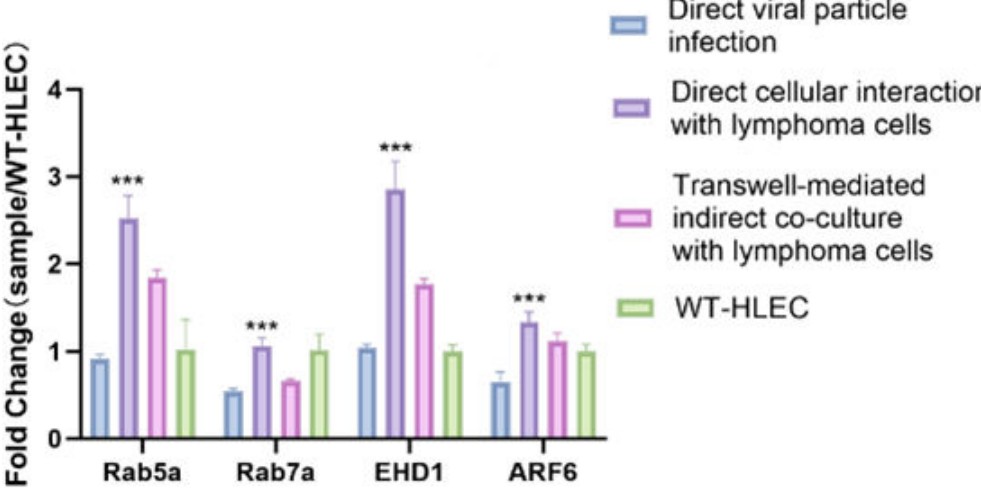

**FIG 6** RT-qPCR detection of endocytosis-related gene expression in HLECs under three different infection modes. ***, $P <$ 0.001. WT, wild type.

## Live-cell imaging and endocytosis protein analysis of HLEC interaction with Akata-GFP/Raji-GFP

Live-cell imaging revealed that HLECs internalized Akata- GFP cells (Fig. 5A) and Raji-GFP cells (Fig. 5B) within 12 h of coculture, forming characteristic "cell-in-cell" structures with emerging weak fluorescence. Progressive intensification of fluorescent signals between 12 and 24 or 72 h confirmed successful EBV infection of HLECs through direct B-lymphoma cell contact (Fig. 5).

## RT-qPCR detection of endocytosis-related gene expression in HLECs

To molecularly characterize the endocytic process in HLECs, we collected HLEC RNAs after complement-dependent cytotoxicity assays in three different modes of infection to examine the expression levels of endocytosis-associated proteins. The results showed that the expression of Rab5a, Rab7a, EHD1, and ARF6 was significantly upregulated in the HLECs exposed to direct Akata cell contact compared to the other two EBV infection strategies, and the difference was statistically significant ($P <$ 0.001, Fig. 6). This suggests that B lymphocyte-mediated EBV dissemination specifically activates the endocytic machinery of endothelial cells.

## Single-cell transcriptomic profiling of EBV-positive nasopharyngeal carcinoma microenvironment

Baseline information for the 24-case sample in this section can be found in the literature (43). Comprehensive single-cell RNA sequencing analysis of 16 EBV-positive NPC specimens and 8 normal nasopharyngeal controls (Fig. 7A) systematically characterized the tumor ecosystem, identifying major cellular subsets through lineage-specific markers: T lymphocytes (CD2/CD3/CD4/CD8), myeloid populations (CD14/CD68), B lymphocytes (CD79/MS4A1), plasmacytoid dendritic cells (CXCR3/IL3RA), fibroblasts (COL1A1/COL1A2), vascular endothelial cells (VWF/PECAM1), and epithelial compartments (EPCAM/KRT19) (Fig. 7B). Spatial transcriptomic integration with single-cell data enabled precise cellular localization and EBV infection status determination, revealing distinct viral tropism with infection probabilities of 18.81% in neoplastic epithelial cells, 10.60% in fibroblasts, 15.74% in macrophages, 8.11% in endothelial cells, 5.71% in naive B cells, 6.01% in plasma cells, and 4.62% in CD4+ T cells, while CD8+ T cells and germinal center B cells remained completely EBV negative (Fig. 7C and D), highlighting the selective cellular permissiveness to EBV infection within the nasopharyngeal tumor microenvironment.

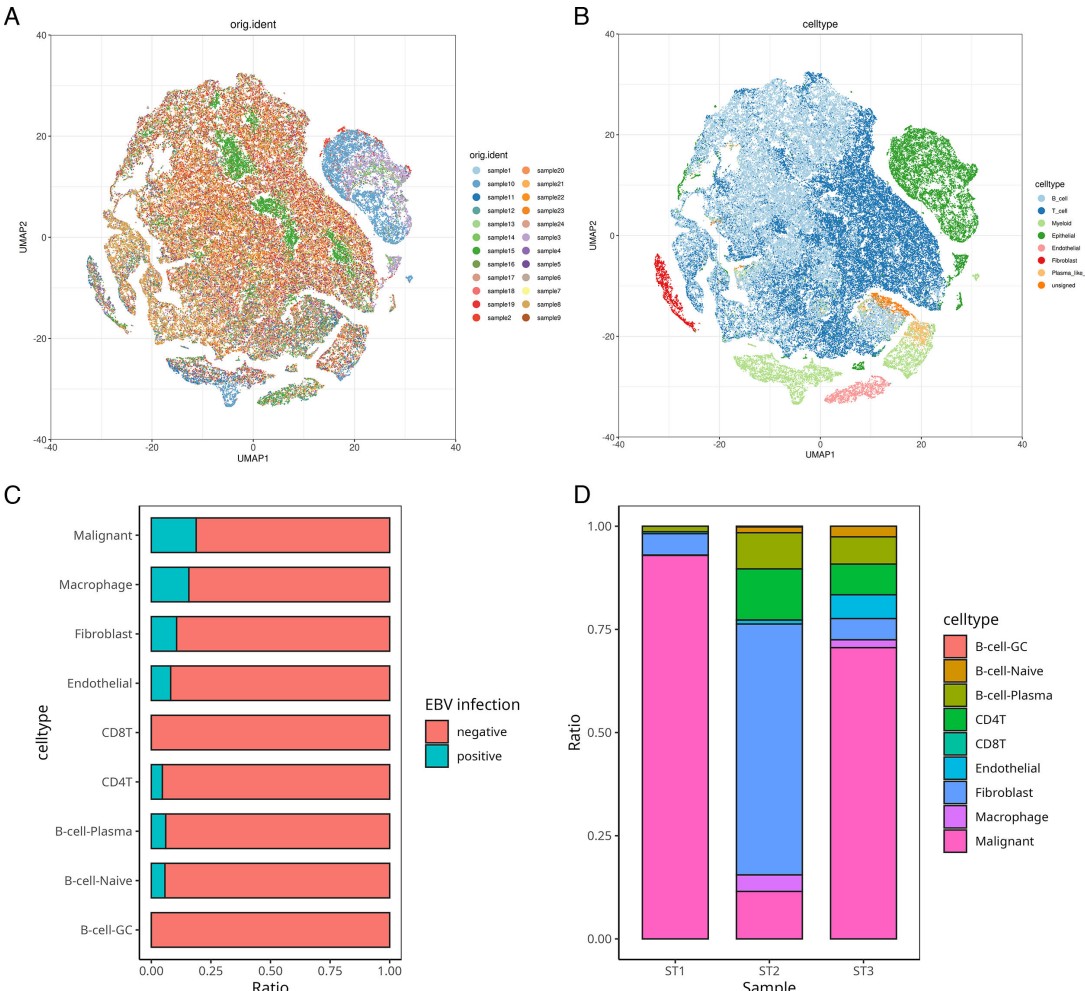

**FIG 7** t-distributed stochastic neighbor embedding (t-SNE) visualization of single-cell data and cell proportion analysis in spatial transcriptomics. (A and B) Results of t-SNE dimensionality reduction visualization of single-cell data. (A) Distribution of cells from different patient sources in reduced dimensional space. (B) Distribution of different cell types in reduced dimensional space. (C and D) Results of cell proportion analysis based on spatial transcriptome (stereo-seq) data. (C) The proportion of EBV-positive and EBV-negative cells in each cell type. (D) The proportion of each cell type in different samples.

## Transmission electron microscopy analysis of EBV infection

This study employed EBV-positive B lymphoma Raji cells to investigate the direct contact-mediated infection mechanism, with transmission electron microscopy revealing distinct cell-in-cell structures following 48 h coculture with HLECs. Electron micrographs demonstrated the presence of Raji cell fragments within HLECs (Fig. 8), providing ultrastructural evidence that EBV may infect HLECs through B lymphocyte-mediated direct cellular interactions. These findings conclusively validate the cell-to-cell transmission pathway of EBV infection in endothelial cells.

## DISCUSSION

NPC is a regionally prevalent malignancy with a high incidence in East and Southeast Asia, and latent EBV infection has been shown to play a crucial role in NPC development (2, 3). EBV is the most common latently infectious virus in humans, with approximately 95% of the global population harboring a latent infection throughout their lifetime; however, no effective vaccine exists to prevent this infection (44, 45). Previous studies have established that B lymphocytes and epithelial cells are susceptible to EBV, with well-characterized infection processes and key receptors (27). Despite their

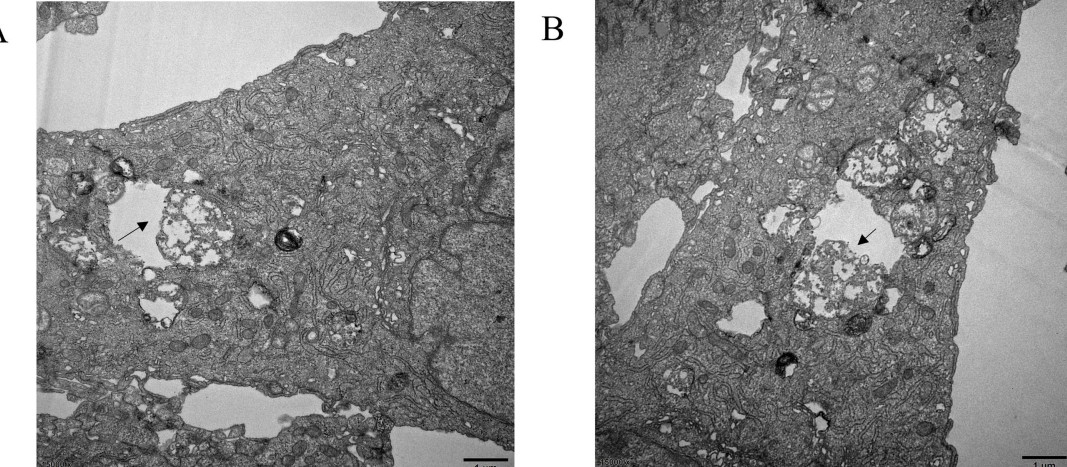

**FIG 8** Transmission electron micrographs demonstrating Raji cell internalization by HLECs (magnification ×15,000; scale bar = 1 µm). (A and B) Representative image showing debris of Raji cell (arrow) within HLECs. HLEC, human lymphatic endothelial cell.

importance within the tumor microenvironment, the role of EBV-infected endothelial cells in facilitating primary tumor immune evasion and promoting distant metastasis has been underexplored, mainly due to the relatively low frequency of these events and the subtle vascular structures in pathological *in situ* hybridization-stained sections.

This study investigated EBV infection in endothelial cells within peritumoral tissues by combining H&E staining of serial sections with *in situ* hybridization. We analyzed the pathological results of basic clinical features. We observed that, among EBV-positive patients, those with EBV-positive endothelial cells exhibited higher N stage, M stage, and clinical stages, as well as an increased risk of recurrence and mortality. These observations suggest that EBV infection in endothelial cells may be a previously underappreciated factor in tumor progression, which contributes to distant metastasis in NPC and is associated with a poorer prognosis. In primary EBV infections, the number of NK cells usually significantly increases (46, 47). In the present study, the detection of cellular immunity indicators revealed a rise in NK cells and a decrease in the proportion of B cells in the endothelial EBER-positive group. Studies have shown that NK cells are crucial in controlling primary EBV infection by clearing infected B cells and releasing immunomodulatory cytokines to enhance antigen-specific T-cell responses, which rationally explains our results (47). However, the exact impact requires expanding the number of cases and more in-depth studies.

The mechanism of EBV infection of different cell types has been well documented in B lymphocytes and epithelial cells. In B cells, EBV entry occurs primarily through the interaction of the viral glycoprotein gp350 with the cell receptor CD21, followed by the interaction of gp42 with major histocompatibility complex class II molecules, promoting membrane fusion and viral entry (48). For epithelial cells, EBV infection begins with the binding of the viral protein BMRF2 to epithelial cell surface integrins, followed by gH/gL glycoprotein interactions with integrins and Ephrin receptor A2, triggering gB activation and fusion of the viral envelope with the epithelial cell membrane (49). Additionally, epithelial cells can be infected through direct contact with EBV-positive B lymphocytes or by endocytosis of B cells, forming cell-in-cell structures (50, 51). Once inside the cell, the EBV capsid is lysed, and the viral genome is transported to the nucleus to initiate replication; after primary lytic infection is controlled, EBV establishes lifelong latency in memory B cells (52).

Although EBV infection of endothelial cells has been observed in various pathological specimens (30–33, 53), the mechanisms of endothelial infection are not well understood. Reactivating latently infected viruses may be a key driving factor for EBV infection in endothelial cells (29, 54, 55). In this study, we propose and validate three potential

pathways for EBV infection of HLECs combined with bioinformatics to determine the proportion of EBV-positive endothelial cells in patients with NPC. We propose that HLECs may be infected by EBV through the following mechanism: intranuclear endocytosis mediated by EBV-positive B lymphocytes or their surface molecules, forming an "intracellular cell" structure. It is worth noting that in addition to the upregulation of EBV-related gene expression in HLECs after EBV infection, we also detected an upregulation of endocytosis-related gene expression. Rab5a (56) and Rab7a (57) regulate early and late endocytic compartments, respectively, while EHD1 (58) and ARF6 (59, 60) are involved in recycling and clathrin-independent endocytosis. These processes are critical for maintaining cellular homeostasis, regulating signaling, and enabling cell migration. It has been reported that EBV infection in human brain microvascular endothelial cells induces endothelial activation and release of inflammatory factors such as TNFα and CCL2, which increase blood-brain barrier permeability and upregulate endothelial adhesion (53, 61). This suggests that EBV infection of endothelial cells may promote regional lymph node and distant metastasis in NPC by disrupting endothelial integrity, increasing permeability, and enhancing interendothelial adhesion. Furthermore, NPC may transfer EBV non-coding RNAs, such as EBERs, to endothelial cells via exosomal pathways, promoting angiogenesis and subsequently increasing peri- and intratumoral microvessel density (62).

Some limitations should be acknowledged in this study. Using consecutive section H&E staining to identify the presence of EBV-infected endothelial cells presents practical challenges, including the need for highly consistent sample processing and relatively long comparison and analysis times. In addition, the functional consequences of EBV-infected endothelial cells need to be further elucidated using *in vitro* and *in vivo* models. Despite these limitations, this study is the first exploring the association of EBV-infected endothelial cells in the tumor microenvironment with NPC progression and prognosis.

## Conclusion

This study demonstrates that EBV may infect vascular endothelial cells through lymphocyte-mediated endocytosis, establishing a novel cell-to-cell transmission pathway that contributes to viral persistence within the tumor microenvironment. Endothelial EBV infection may serve as a critical mediator of tumor progression, immune evasion, and disease recurrence in NPC. These findings provide mechanistic insights into EBV pathogenesis and highlight endothelial cells as potential therapeutic targets for preventing NPC development and relapse.

## ACKNOWLEDGMENTS

Thanks to the Key Laboratory of Early Prevention and Treatment for Regional High Frequency Tumor (Guangxi Medical University), Ministry of Education, Life Sciences Institute of Guangxi Medical University, and Department of Head and Neck Tumor Surgery, Affiliated Tumor Hospital of Guangxi Medical University, for their technical support. Sincere thanks to all those who have contributed to this study.

This study was supported by the National Natural Science Foundation of China (No. 82160386), the Guangxi Natural Science Foundation of China (Nos. 2023GXNSFAA026189, 2024GXNSFDA010032, and 2021GXNSFAA075042), and the Guangxi Science and Technology Program (AD25069077).

X.C.: validation, formal analysis, investigation, writing of the original draft, and visualization; Z.H.: methodology, validation, formal analysis, investigation, and writing of the original draft; X.H.: formal analysis, writing of the original draft, and visualization; X.Y.: project administration; K.-l.P.: validation; X.-l.H.: investigation; C-j.L.: validation; Z.W.: conceptualization, resources, supervision, methodology, project administration, writing (review and editing), and funding acquisition; Y.X.: conceptualization, methodology, resources, writing (review and editing), supervision, project administration, and funding acquisition.

## AUTHOR AFFILIATIONS

[1]Key Laboratory of Early Prevention and Treatment for Regional High Frequency Tumor (Guangxi Medical University), Ministry of Education, Guangxi, China

[2]School of Life Science and Medical Engineering, Guangxi Medical University, Guangxi, China

[3]Department of Head and Neck Tumor Surgery, Affiliated Tumor Hospital of Guangxi Medical University, Guangxi, China

[4]School of International Education, Guangxi Medical University, Guangxi, China

## AUTHOR ORCIDs

Xu-lin Chen  http://orcid.org/0009-0005-9101-5324

Zheng-bo Wei  http://orcid.org/0000-0001-7858-4796

Ying Xie  http://orcid.org/0000-0002-8856-4243

## FUNDING

| Funder | Grant(s) | Author(s) |
| --- | --- | --- |
| National Natural Science Foundation of China | No.82160386 | Ying Xie |
| Natural Science Foundation of Guangxi Zhuang Autonomous Region | 2023GXNSFAA026189 | Zheng-bo Wei |
| Natural Science Foundation of Guangxi Zhuang Autonomous Region | 2024GXNSFDA010032 | Ying Xie |
| Scientific Research and Technology Development Program of Guangxi Zhuang Autonomous Region (Guangxi Science and Technology Planning Project) | AD25069077 | Ying Xie |

## AUTHOR CONTRIBUTIONS

Xu-lin Chen, Formal analysis, Investigation, Validation, Visualization, Writing – original draft | Zhong-heng Huang, Formal analysis, Investigation, Methodology, Validation, Writing – original draft | Xiu-han Huang, Formal analysis, Visualization, Writing – original draft | Xi Yao, Project administration | Ke-ling Pang, Validation | Xin-lu He, Investigation | Cui-juan Luo, Validation | Zheng-bo Wei, Conceptualization, Funding acquisition, Methodology, Project administration, Resources, Supervision, Writing – review and editing | Ying Xie, Conceptualization, Funding acquisition, Methodology, Project administration, Resources, Supervision, Validation, Writing – review and editing

## DATA AVAILABILITY

The data sets analyzed in the current study are available from GEO (https://www.ncbi.nlm.nih.gov/geo/).

## ETHICS APPROVAL

The study protocol obtained approval from both the institutional ethics committee of the Affiliated Tumor Hospital and the independent Ethics Committee of Guangxi Medical University (Ethics No. 20210046), with written informed consent obtained from all participants.

## ADDITIONAL FILES

The following material is available online.

## Open Peer Review

**PEER REVIEW HISTORY (review-history.pdf).** An accounting of the reviewer comments and feedback.

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
