## [Reviewer comments · Microbiology Spectrum]

Microbiology Spectrum

Epstein-Barr Virus (EBV) infection of endothelial cells via endocytosis is associated with a poor prognosis in nasopharyngeal carcinoma.

Xu-lin Chen, Zhong-heng Huang, Xiu-han Huang, Xi Yao, Ke-ling Pang, Xin-lu He, Cui-juan Luo, Zheng-bo Wei, and Ying Xie

Corresponding Author(s): Ying Xie, Guangxi Medical University

Review Timeline:

Submission Date:	April 21, 2025
Editorial Decision:	April 23, 2025
Revision Received:	April 25, 2025
Accepted:	May 25, 2025

Editor: Tao Deng

Reviewer(s): The reviewers have opted to remain anonymous.

Transaction Report:

DOI: <https://doi.org/10.1128/spectrum.03427-24>

Re: Spectrum03427-24 (**Epstein-Barr Virus (EBV) infection of endothelial cells via endocytosis is associated with a poor prognosis in nasopharyngeal carcinoma.**)

Dear Dr. Ying Xie:

Thank you for submitting your manuscript to Microbiology Spectrum. Please revise your point-by-point response to the reviewer's comments by clearly indicating the specific changes made in the manuscript. Be sure to include explicit references to the relevant line numbers, as well as any newly added figures or tables. The current response is difficult to follow and does not clearly show how the manuscript has been improved based on the reviewer's feedback.

Please return the manuscript within 30 days; if you cannot complete the modification within this time period, please contact me. If you do not wish to modify the manuscript and prefer to submit it to another journal, notify me immediately so that the manuscript may be formally withdrawn from consideration by Spectrum.

Revision Guidelines

Sincerely,
Tao Deng
Editor
Microbiology Spectrum

Re: Spectrum03427-24R1 (**Epstein-Barr Virus (EBV) infection of endothelial cells via endocytosis is associated with a poor prognosis in nasopharyngeal carcinoma.**)

Dear Dr. Ying Xie:

Your manuscript has been accepted, and I am forwarding it to the ASM production staff for publication. Your paper will first be checked to make sure all elements meet the technical requirements. ASM staff will contact you if anything needs to be revised before copyediting and production can begin. Otherwise, you will be notified when your proofs are ready to be viewed.

Sincerely,
Tao Deng
Editor
Microbiology Spectrum